# Value Function Decomposition for Iterative Design of Reinforcement Learning Agents

**James MacGlashan**[†]
james.macglashan@sony.com

**Evan Archer**[†*]
evan.archer@sony.com

**Alisa Devlic**[†*]
alisa.devlic@sony.com

**Takuma Seno**[†*]
takuma.seno@sony.com

**Craig Sherstan**[†*]
craig.sherstan@sony.com

**Peter R. Wurman**[†]
peter.wurman@sony.com

**Peter Stone**[†‡]
pstone@cs.utexas.edu

## Abstract

Designing reinforcement learning (RL) agents is typically a difficult process that requires numerous design iterations. Learning can fail for a multitude of reasons, and standard RL methods provide too few tools to provide insight into the exact cause. In this paper, we show how to integrate *value decomposition* into a broad class of actor-critic algorithms and use it to assist in the iterative agent-design process. Value decomposition separates a reward function into distinct components and learns value estimates for each. These value estimates provide insight into an agent's learning and decision-making process and enable new training methods to mitigate common problems. As a demonstration, we introduce SAC-D, a variant of soft actor-critic (SAC) adapted for value decomposition. SAC-D maintains similar performance to SAC, while learning a larger set of value predictions. We also introduce decomposition-based tools that exploit this information, including a new reward *influence* metric, which measures each reward component's effect on agent decision-making. Using these tools, we provide several demonstrations of decomposition's use in identifying and addressing problems in the design of both environments and agents. Value decomposition is broadly applicable and easy to incorporate into existing algorithms and workflows, making it a powerful tool in an RL practitioner's toolbox.

## 1 Introduction

Deep reinforcement-learning (RL) approaches have achieved successes in a range of application areas such as gaming ([5, 32, 36, 38]), robotics ([22]), and the natural sciences ([25, 35]). Despite these successes, applying RL techniques to complex control problems remains a daunting undertaking, where initial attempts often result in underwhelming performance. Unfortunately, there are many reasons why an agent may fail to learn a good policy, making it difficult to diagnose which reason(s) caused a particular agent to fail. For example: an agent may fail because the state features were insufficient to make accurate predictions, different task objectives defining the reward function may be imbalanced, the agent may fail to sufficiently explore the state-action space, values may not accurately propagate to more distant states, the neural network may not have sufficient capacity to

---

[†]Sony AI

[*]Equal contribution

[‡]The University of Texas at Austin

36th Conference on Neural Information Processing Systems (NeurIPS 2022).

approximate the policy or value function(s), or, there may be subtle differences between training and evaluation environments. Without a way to diagnose the causes of poor performance or to recognize when a problem has been remedied, practitioners typically engage in a long trial-and-error design process until an agent reaches a desired level of performance. Frustrations with this trial-and-error process have been expressed in other work [16].

We describe how *value decomposition*, a simple, broadly-applicable technique, can address these application challenges. In RL, the agent receives a reward that is often a sum of many reward components, each designed to encode some aspect of the desired agent behavior. From this *composite reward*, it learns a single *composite value function*. Using value decomposition, an agent learns a *component value function* for each reward component. To perform policy optimization, the composite value function is recovered by taking a weighted sum of the component value functions. While prior work has proposed value decomposition methods for discrete-action Q-learning [19, 29, 31], we show how value decomposition can be incorporated into a broad class of actor-critic (AC) methods. In addition, we introduce SAC-D, a version of soft actor-critic (SAC) [13, 14] with value decomposition, and explore its use in multi-dimensional continuous-action environments. We also introduce the *influence* metric, which measures how much an agent's decisions are affected by each reward component.

While earlier work focuses on its use in reward design [16, 19], value decomposition can facilitate diagnosis of a wide range of issues and enable new training methodologies. To demonstrate its utility, in Sec. 6 we show how to use it to: (1) diagnose insufficient state features; (2) diagnose value prediction errors and exploit the decomposed structure to inject background-knowledge; and (3) identify reward components that are inhibiting exploration and mitigate the effect by gradually incorporating component predictions into policy optimization.

Value decomposition's additional diagnostic and training capabilities come at the cost of a more-challenging prediction problem: instead of learning a single value function, many must be learned. To investigate if this difficulty negatively impacts agent performance, we compare the average performance of SAC-D to SAC on benchmark environments. We find that a naive implementation of SAC-D underperforms SAC and then show how to improve SAC-D so that it matches and sometimes exceeds SAC's performance. These improvements may also be applied to value decomposition for other AC algorithms.

While variations of value decomposition has been explored extensively in past work (see Sec. 7), in this paper, we make the following contributions. (1) We show how to integrate value decomposition into a broad class of actor-critic algorithms. (2) We analyze the performance of different implementations of value decomposition for SAC on a range of benchmark continuous-action environments. (3) We introduce the *influence* metric: a novel value decomposition metric for measuring how much each reward component affects decision-making. (4) We provide a set of illustrative examples of how value decomposition and influence can be used to diagnose various kinds of learning challenges. (5) We describe new training methods that exploit the value decomposition structure and can be used to mitigate different learning challenges.

## 2 Background

### 2.1 MDPs and Q-functions

In RL, an agent's interaction with the environment is modeled as a Markov Decision Process (MDP): $(S, A, P, R, \gamma)$, where $S$ is a set of states, $A$ is a set of actions, $P : S \times A \times S \to \mathbb{R}$ is a state transition probability function $P(s, a, s') = \Pr(S_{t+1} = s' | S_t = s, A_t = a)$, $R : S \times A \to \mathbb{R}$ is a reward function $R(s, a) = E[R_{t+1} | S_t = s, A_t = a]$, and $\gamma \in [0, 1]$ discounts future rewards.[4] The goal of an agent is to learn a policy $\pi(a|s)$ that maps states to an action probability distribution that maximizes the sum of future rewards. The agent is trained to maximize the discounted return $E\left[\sum_t^\infty \gamma^t R(s_t, a_t)\right]$. The $Q$-function maps state-action pairs to the expected cumulative discounted reward when starting in state $s$, taking action $a$, and then following policy $\pi$ thereafter:

$$Q^\pi(s, a) \triangleq \mathbb{E}\left[\sum_{t=0}^\infty \gamma^t R(s_t, a_t) | \pi, s_0 = s, a_0 = a\right]. \tag{1}$$

---

[4]For continuing tasks, $\gamma$ must be $< 1$, and we only consider algorithms for $\gamma < 1$.

## 2.2 Soft actor-critic

Soft actor-critic (SAC) [13, 14] is an off-policy actor-critic algorithm parameterized with five neural networks: a stochastic policy network $\pi$ with parameters $\phi$, and two pairs of $Q$-functions and target $Q$-functions with parameters $(\theta_1, \theta_2)$ and $(\bar{\theta}_1, \bar{\theta}_2)$, respectively. As with other actor-critic algorithms, SAC has two main steps: policy evaluation (in which it estimates the Q-function for policy $\pi$), and policy improvement (in which it optimize the policy to maximize its Q-function estimates). Unlike other actor-critic algorithms, SAC optimizes a maximum entropy formulation of the MDP, in which rewards are augmented with policy entropy bonuses that prevent premature policy collapse. To perform policy evaluation and improvement, SAC minimizes the following loss functions simultaneously:

$$L_{Q_i} = \mathbb{E}\left[\frac{1}{2}\left(Q(s,a;\theta_i) - y\right)^2\right] \text{ for } i \in \{1, 2\}, \tag{2a}$$

$$L_\pi = \mathbb{E}\left[\alpha \log \pi(u|s;\phi) - \min_{j \in \{1,2\}} Q(s,u;\theta_j)\right], \tag{2b}$$

where $(s, a, r, s')$ transitions are drawn from an experience replay buffer, $y := r + \gamma\left(\min_{j \in \{1,2\}} Q(s', a'; \bar{\theta}_j) - \alpha \log \pi(a'|s';\phi)\right)$, $a' \sim \pi(\cdot|s';\phi)$, $u \sim \pi(\cdot|s;\phi)$, and $\alpha$ is an (optionally learned) entropy regularization parameter. The $\min$ of $Q$-function pairs addresses overestimation bias in value function estimation [11, 15]. The parameters $\bar{\theta}_1$ and $\bar{\theta}_2$ are updated toward $\theta_1$ and $\theta_2$ via an exponentially moving average each step.

## 2.3 Environments

Throughout the paper, we italicize descriptive names for the components of the continuous-action Lunar Lander (LL), Bipedal Walker (BW) and Bipedal Walker Hardcore (BWH) environments. In LL, an agent must land a spacecraft in the center of a landing zone using as little fuel as possible. The reward components include: a reward for successful *landing*; penalties for crashing (*crash*) and engine usage (*main*, *side*); shaping rewards used to encourage the agent to stay upright (*angle*), move towards the center of the landing pad (*position*) with low velocity (*velocity*) and land with both legs (*right leg*, *left leg*). In BW, an agent learns to make a 2-legged robot walk. The reward components include: a reward for *forward* progress, a penalty for falling (*failure*), a cost for actions (*control*), and a shaping reward to discourage head movement (*head*). BWH is identical to BW, but adds additional obstacles for the agent to navigate.

## 3 Value decomposition for actor-critic methods

Most RL algorithms estimate the value function and use it to improve its policy. Unfortunately, value functions and policies provide little insight into the agent's decision-making. However, reward functions are often *composite* functions of multiple *component* state-action signals. By learning a value function estimate for the current policy for each component, practitioners gain insight into what the agent expects to happen and how these reward components interact. Naturally, policy improvement still requires the composite value function. In Sec. 3.1 we show how the composite Q-function can be recovered from the component Q-functions. From this property, a range of actor-critic algorithms can be adapted to use value decomposition by following the below template.[5]

1. Alter Q-function networks to have $m$ outputs instead of 1, where $m$ is the number of reward components.
2. Use the base algorithm's Q-function update for each of the $m$ components, replacing the composite reward term with the respective component reward term.
3. Apply the base algorithm's policy improvement step by first recovering the composite Q-function.

For example, this template can be applied to algorithms that use TD(0) [33], or ones that use Retrace [27]. It works with algorithms that improve the policy by differentiating through the Q-function [11, 14, 23], and ones that fit it toward non-parametric target action distributions [1].

---

[5]Composite state value functions can similarly be recovered; however, Q-functions allow for deeper introspection, so we focus on that setting in this work.

---

**Algorithm 1** SAC-D and SAC-D-CAGrad Update

---

**Require:** Experience replay buffer $B$; twin $Q$-function parameters $\theta_1, \theta_2$ (with $\Theta = \theta_1 \cup \theta_2$) and target parameters $\bar{\theta}_1, \bar{\theta}_2$; policy parameters $\phi$; discount factor $\gamma$; entropy parameter $\alpha$; reward weights $w \in \mathbb{R}^{m+1}$; learning rates $\lambda_q, \lambda_\pi$; target network step size $\eta$; Boolean use_cagrad for SAC-D-CAGRAD or SAC-D.
 1: Sample transition (minibatch) $(s, a, r, s') \sim B$           $\triangleright$ $r \in \mathbb{R}^m$ is a vector of $m$ reward components
 2: Sample policy actions $a' \sim \pi(\cdot|s'; \phi)$ and $u \sim \pi(\cdot|s; \phi)$
 3: $r_{m+1} \leftarrow \gamma \alpha \log \pi(a'|s'; \phi)$                $\triangleright$ Extend reward vector to include entropy reward
 4: $j \leftarrow \underset{j \in \{1,2\}}{\arg \min} \sum_i^{m+1} w_i Q_i(s', a'; \bar{\theta}_j)$       $\triangleright$ Find target network by minimum composite $Q$-value
 5: $y_i \leftarrow r_i + \gamma Q_i(s', a'; \bar{\theta}_j)$
 6: $LQ_i \leftarrow \sum_{j=1}^2 \frac{1}{2}\left(Q_i(s, a; \theta_j) - y_i\right)^2$
 7: $L\pi \leftarrow \alpha \log \pi(u|s; \phi) - \min_{j \in \{1,2\}} \sum_i^{m+1} w_i Q_i(s, u; \theta_j)$
 8: **if** use_cagrad **then**
 9:     $\Theta \leftarrow \Theta - \lambda_q \text{CAGRAD}(\mathbf{J}_{LQ}, \Theta)$
10: **else**
11:     $\Theta \leftarrow \Theta - \lambda_q \nabla_\Theta \frac{1}{m+1} \sum_i^{m+1} LQ_i$
12: **end if**
13: $\phi \leftarrow \phi - \lambda_\pi \nabla_\phi L\pi$
14: Update target networks $\bar{\Theta} \leftarrow (1 - \eta)\bar{\Theta} + \eta\Theta$

---

Although this template is conceptually simple, learning component $Q$-functions poses a more difficult prediction problem: multiple predictions must be learned instead of one composite prediction. Ideally, this increased difficulty would not negatively impact agent performance. In Sec. 3.2 we introduce SAC-D, an adaptation of SAC to use value decomposition, and describe additions we made to the above template to maintain performance with conventional SAC. Although these additions are contextualised to SAC, they are general and can be used when adapting other actor-critic algorithms.

## 3.1 Recovering the composite Q-function

We assume the environment's reward function is a linear combination of $m$ components: $R(s, a) \triangleq \sum_i^m w_i R_i(s, a)$, where $w_i \in \mathbb{R}$ is a scalar *component weight* for the $i$th component and $R_i(s, a) \to \mathbb{R}$ is the reward function of the $i$th component for state-action pair $(s, a)$. Applying the linearity of expectation, we find the $Q$-function inherits the linear structure from the reward[6]:

$$Q^\pi(s, a) = \sum_i^m w_i Q_i^\pi(s, a), \tag{3}$$

where we define the $i$'th *component Q-function* as $Q_i^\pi(s, a) \triangleq E[\sum_t \gamma^t R_i(s_t, a_t)|\pi, s_0 = s, a_0 = a]$. Unless otherwise specified, we assume $w_i = 1$ for all $i$. Because the component weights are factored out of the component Q-functions, they may be varied without changing the component prediction target, allowing the policy to be evaluated for any weight combination. Although the assumption of linearity may seem restrictive, note that each reward component may be a non-linear function of state variables, allowing for very expressive environment rewards. Furthermore, many environments, including all the environments we investigate in this paper, are naturally structured as a sum of (non-linear) reward components.

## 3.2 SAC with value decomposition

Here we introduce SAC-D (Alg. 1), an adaptation of SAC to use value decomposition. Adapting SAC only requires one additional consideration from our template: the entropy bonus reward term SAC adds is treated as an $m + 1$'th reward component: $R_{m+1}(s') \triangleq \gamma \alpha \log \pi(a'|s'; \phi)$ (line 3). However, this approach, which we refer to as *SAC-D-Naive*, underperforms SAC in many settings. We found two additional modifications essential to match the performance of SAC. The first concerns how we apply the twin-network minimum of Eq. 2 in the context of value decomposition. The second is to use **C**onflict-**A**verse **Grad**ient descent (CAGrad) [24] to address optimization problems that arise when training multi-headed neural networks. We refer to SAC with value decomposition and the

---

[6]See Theorem A.1 for the proof. This linear decomposition property of value functions has been explored elsewhere [3, 9], but in different contexts and with different motivations. See Sec. 7 for more information.

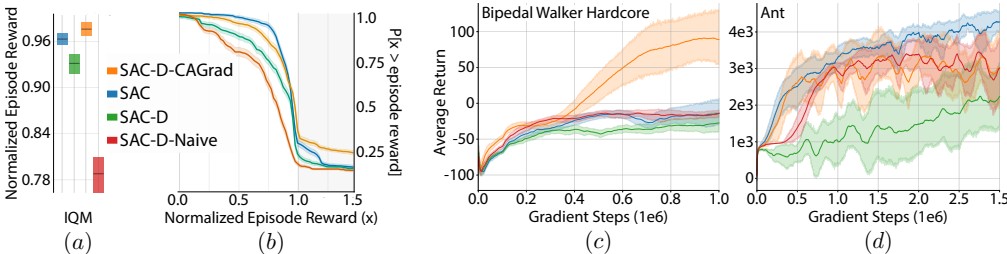

Figure 1: **Benchmark results (a)-(b)** Comparison of algorithms over 8 environments. For each algorithm, scores are collected from the 10 final checkpoints (10 episodes × 10 final policies), and normalized on individual environments. The 95% CIs are estimated using stratified bootstrap sampling. **(a)** The interquartile mean (IQM), which gives the mean over the central 50% of values. **(b)** For each normalized score $x$, % of runs achieving a score of at least $x$. Grey region, $x > 1.0$, corresponds to runs outperforming the mean SAC baseline score. **(c)** Training curves on continuous-control benchmarks for BWH. Solid lines represent mean episode return over 10 trials. Shaded areas are confidence intervals. Lines are uniformly smoothed for clarity. **(d)** Same as **(c)**, but for Ant.

twin-network correction as *SAC-D*, and the variant with twin-network correction and CAGrad as *SAC-D-CAGrad*.

**Twin-network minimums in value decomposition:** In SAC, the Q-value target is the minimum of two Q-function networks (Eq. 2). Using the same Q-value update rule for each component, as described in our template, suggests using a minimum for each component target: $q_i := \min_{j \in \{1,2\}} Q_i(s, a; \bar{\theta}_j)$. However, this is not a good choice in practice. The purpose of the twin-network minimum is to mitigate overestimation bias from the feedback loop of the policy optimizing the Q-function. Because the policy optimizes the composite $Q$-function, a better approach is to use all the predictions from the network with the minimum composite Q-function (Alg. 1, lines 5-6). This approach reduces underestimation bias and improves performance compared to an element-wise minimum (see Sec. 4).

**Mediating the difficulty of multi-objective optimization:** Even though the scalar values of the composite $Q$-function are identical to those used in SAC, simultaneous optimization of all $Q_i^\pi$ components may introduce training problems common in multi-objective optimization: conflicting gradients, high curvature and large differences in gradient magnitudes [39].[7] The CAGrad method, designed for the multi-task RL setting, addresses these issues by replacing the gradient of a multi-task objective with a weighted sum of per-task loss gradients. This updated gradient step maximizes the improvement of the worst-performing task on each optimization step, and still converges to a minimum of the unmodified loss. We incorporate CAGrad into SAC-D by treating each component as a "task", and update the gradient vector accordingly (Alg. 1, line 9).

## 4 Robustness experiments

We benchmark SAC-D, SAC-D-CAGrad and SAC-D-Naive against SAC on a selection of continuous-action Gym [7] environments. For each environment, we exposed existing additive reward components *without* altering the behavior of the environments or their composite rewards. That is, these environments already implemented their reward functions as a linear combination of separate reward components and we simply exposed that information to the algorithm (for details, see App. B). As outlined in App. C, we used hyperparameters previously published for use with SAC [14] for all experiments. We tied SAC-D's hyperparameters to SAC's because our goal is for value decomposition to be a drop-in addition without significant loss in agent-performance. However, it is possible better performance could be reached with tuning.

Figure 1(a) shows the performance of each algorithm aggregated across all environments and all experimental runs. Figure 1(b) shows the same information, but highlights the distribution of scores across experimental runs. In aggregate, SAC-D-CAGrad slightly outperforms SAC, although it has a broader range of performance scores. SAC-D-Naive significantly underperforms SAC.

---

[7]Multi-headed prediction can also improve representation learning, as in work on auxiliary tasks [18, 26]

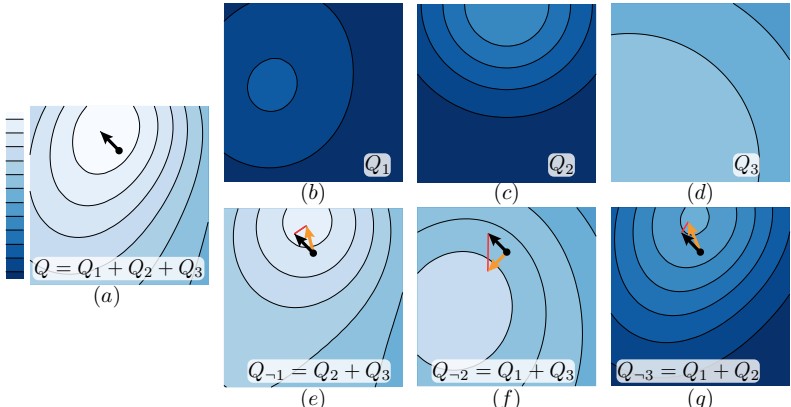

Figure 2: **Influence calculation for a three-component environment with a 2D action space.** In this illustrative example, component 2 has high influence, while components 1 and 3 have low influence, but for different reasons. Each figure shows a contour-map view of the Q-function over the 2D action space for the same state. **(a)** The composite Q-function the policy optimizes. The black point is a (near-optimal) policy selection, and the black arrow is the policy gradient. **(b)-(d)** Each component Q-function. **(e)-(g)** Component-ablated composite Q-function surfaces $Q_{\neg i}$ used to compute influence for each component. Influence is defined by the length of the red line, which is the difference of the policy gradients computed for the composite Q-function $Q$ (black arrow) and for $Q_{\neg i}$ (orange arrow). **(e)** Component 1 has low influence because it is out-weighed by component 2. **(f)** Conversely, component 2 has high influence; without it, the policy would move toward the peak of component 1. **(g)** While component 3 contributes a large value, it has low influence because the value is nearly uniform; much of its value is received regardless of the action selected.

We provide training curves for the 8 environments investigated in App. D, but highlight the atypical training curves for BWH (Fig. 1(c)) and Ant (Fig. 1(d)). In the case of BWH, SAC-D-CAGrad significantly outperforms SAC. It was not the goal of this work for SAC-D to improve on SAC. Rather, we sought to provide more insights into the learning process. As such, we make no strong claims about when SAC-D can be expected to outperform SAC. Nevertheless, this result does suggest that SAC-D may sometimes benefit from auxiliary task learning. We leave this question as a subject for future investigation.

In Ant, SAC-D (without CAGrad) underperforms all other methods. We found that infrequent environment termination causes large Q-function errors and leads to catastrophic gradient conflicts. Further analysis of termination issues and CAGrad behavior is described in Appendices E and F respectively.

## 5   Reward component influence

It can be difficult to understand how an agent's predictions interact to affect decision-making. We now introduce the reward *influence* metric, which indicates how much each component contributes to an agent's decision. Intuitively, low influence means that removing a component would not alter decision-making; high influence means that removing it would significantly alter decision-making.

For multi-dimensional continuous actions, we define the *optimal influence* of component $i$ in state $s$ by how much the optimal policy $\pi^*$ in state $s$ differs from the optimal policy when component $i$ is removed: $\mathcal{I}_i^*(s) \triangleq ||\pi^*(s) - \pi_{\neg i}^*(s)||_2$, where $\pi_{\neg i}^* \triangleq \arg\max_\pi E\left[\sum_t^\infty \gamma^t \sum_{j \neq i} w_j R_j(s_t, a_t)|\pi\right]$.[8]

In practice, we apply two approximations to $\mathcal{I}_i^*(s)$ for computational efficiency. First, rather than compare the difference of optimal policies, we compare the difference between one step of policy improvement: $\mathcal{I}_i^\pi(s) \triangleq || \arg\max_a Q^\pi(s, a) - \arg\max_a Q_{\neg i}^\pi(s, a)||_2$, where $Q_{\neg i}^\pi(s, a) \triangleq \sum_{j \neq i} w_j Q_j^\pi(s, a)$. Second, since $\arg\max$ is computationally demanding and sensitive to statistical noise, we replace the $\arg\max_a Q(s, a)$ policy improvement operator with a policy gradient-step

---

[8]Discrete-action spaces could use a probability distance measure, but we focus on continuous-action spaces.

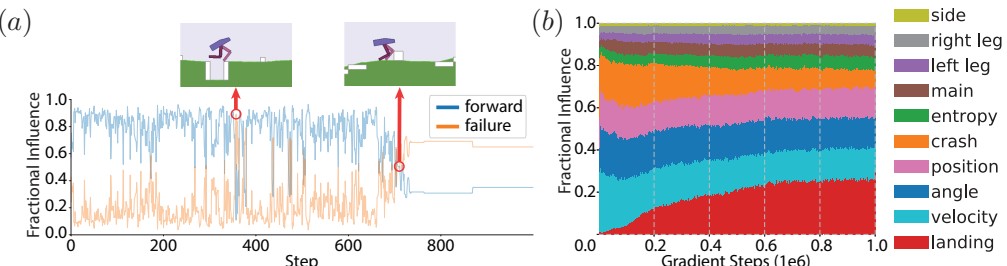

Figure 3: **Fractional influence (a)** Fractional influence for 1 evaluation trajectory of BWH. For this agent and trajectory, the *forward* component dominates the *failure* component in decision-making, except at two points (red arrows). At the first point, the agent was at risk of falling into a pit. After the agent stabilizes, the *forward* component returns to being dominant. After the second point, the agent becomes unable to move forward. *Failure*'s dominance indicates that if the agent cannot move forward, it prefers standing to falling. If it is possible to surmount this obstacle, more exploration may be required. **(b)** Mean stacked fractional influence across 10 trials of Lunar Lander. We observe: (1) the sparse landing reward doesn't meaningfully contribute to decision-making in the earliest part of training; (2) crashing influence starts high and then decreases, suggesting the agent initially learns to avoid crashing; (3) the position, angle, and velocity reward components – each a shaping reward – maintain large influence, suggesting possible overreliance on shaping rewards; (4) numerous reward components never have much influence; they might be unnecessary or require more attention.

operator (typical in RL algorithms like SAC): $\bar{a} + \lambda \nabla_{\bar{a}} Q(s, \bar{a})$, where $\bar{a}$ is a deterministic policy action selection (such as the mode) and $\lambda \in (0, 1)$ is a step size.

When taking the difference of the gradient-step operator applied to the $Q^\pi$ and $Q^\pi_{\neg i}$ surfaces, the $\bar{a}$ terms cancel, and $\lambda$ can be factored out of the norm. The result is the *influence* metric (Fig. 2):

$$I_i^\pi(s; \theta) \triangleq \lambda || \nabla_{\bar{a}} Q^\pi(s, \bar{a}; \theta) - \nabla_{\bar{a}} Q^\pi_{\neg i}(s, \bar{a}; \theta) ||_2. \tag{4}$$

The raw magnitudes of component influence can be informative by themselves; for example, a sharper Q-function surface leads to larger influence values. However, to compare influence values across components, we typically compute the *fractional influence* by normalizing the (always non-negative) influence: $\hat{I}_i^\pi(s; \theta) \triangleq \frac{I_i^\pi(s; \theta)}{\sum_j^m I_j^\pi(s; \theta)}$.

We use several techniques to visualize the fractional influence. For trajectories, plotting influences over timesteps may help identify and explain key decision points (Fig. 3(a)). When studying an agent's behavior across training, we maintain summary statistics of fractional influence. We visualize mean fractional influence across all components as a stack plot, sorted so that the component with the maximum influence at the end of training is at the bottom. Figure 3(b) shows such a diagram for the Lunar Lander environment; we provide figures for all environments in App. G.

# 6 Value decomposition: strategies for agent design

Agent design is often a brute-force process of trial and error: when an agent doesn't perform as expected, we choose some aspect of the agent's design to vary, and then we train again. Although this approach can succeed, it can be expensive in both time and computation.

In this section, we illustrate a different approach, showing how a decomposed reward helps break from trial-and-error by encouraging the designer to consider an agent's point of view. In three examples, each using either the LL or BWH environments (Sec. 2.3), we draw on value decomposition tools to: (1) identify learning problems by comparing components' empirical returns to their predictions; (2) constrain component value estimation; (3) identify adverse reward interactions with the influence metric; (4) dynamically re-weight reward components.

These examples are not just meant to demonstrate these specific techniques, nor to demonstrate significant performance improvements on these well-tested benchmark environments (in fact, SAC-D eventually produces good policies for both environments without additional tuning). Rather, the goal is to showcase a general approach to agent design for novel applications in which iteration and

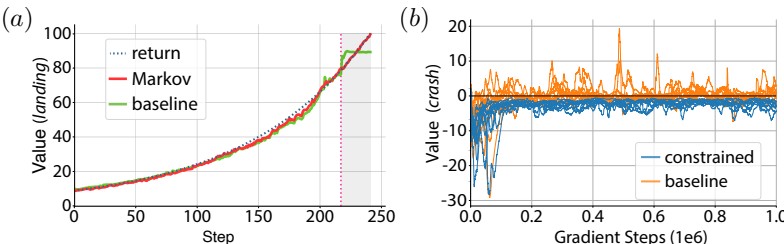

Figure 4: **Remedies for incorrect value predictions.** **(a)** For the *landing* component empirical return (dashed blue), the value prediction (green) for a value function trained with baseline features matches the return well except for a final plateau in the shaded region. Training a value function with the added $V_0^{\text{trace}}$ feature (red) improves predictions in this region. The dashed pink line indicates the point after which we compute error metrics in App. H. **(b)** Value predictions for the *crash* component with sign constraints (blue lines; using both target and prediction constraints) and without (orange lines). Each line represents an independent trial. Unconstrained estimates can be positive; constrained value estimates keep predictions negative.

failure is costly. Combined with statistical tools that allow us to reason with small-sample statistics, value decomposition provides a vocabulary to describe an agent's behavior and interpretable tools for targeted interventions. Even these small examples are more intuitive – and less computationally demanding – than they would be with a trial-and-error approach.

For all examples, training parameters are identical to the robustness results of Sec. 4.

**Diagnosing and improving insufficient features:** We analyze the behavior of an agent trained on Lunar Lander by comparing each component's empirical return to the agent's value predictions, $Q_i$. The agent is trained to land successfully, and generally the component value predictions match their empirical returns well. Curiously, all component predictions are flat near the end of the episode. For most components, these flat predictions are a good match for their returns, but not for *landing*. Investigating the landing dynamics, we found the simulator waits many steps after touchdown before producing a landing reward. During this period, the observations are constant, suggesting the features are inadequate to represent *landing*'s return. To make the observations Markov, we introduce a new feature to the agent's observations that indicates the duration since the agent's velocity (horizontal, vertical and angular) went to zero: $V_0^{\text{trace}}(t) = V_0^{\text{steps}}(t)/c$, where $V_0^{\text{steps}}(t)$ is the number of time steps since all the velocities dropped below a threshold and $c$ is a fixed normalizing constant. With this feature, post-landing predictions show a marked improvement (see Figure 4(a) and Appendices C, H).

**Diagnosing and mitigating value errors using domain knowledge:** Lunar Lander's design makes clear that certain reward components are always non-positive (*crash, main, side*) while others are always non-negative (*landing*). However, we observe that the agent's decomposed value predictions do not always match these bounds. In particular, value predictions of *crash* have a tendency to oscillate about 0 after the agent learns to land. Value decomposition allows us to explicitly enforce a sign constraint on *crash* (Figure 4(b); see App. I for details). In this particular example, constraints do not alter policy learning performance, but the resulting predictions are easier to interpret, and the same technique may improve performance in more complex environments.

**Mitigating an adverse reward with component weight scheduling:** Under the BWH reward function, a random policy is far more likely to experience an unsuccessful outcome (falling over) than a successful one (walking forward). This bias can inhibit agent exploration early in training, causing an agent's policy to fall into a local minimum (the agent stands still). Here, we diagnose this dynamic using component predictions and influence metrics, and remedy it by varying a single component weight during training.

We find that the *failure* component's fractional influence dominates the *forward* component's fractional influence early in training, and that this relationship reverses as agent performance improves (Fig. 5(a)). The *forward* component's near-zero value predictions (Fig. 5(b)) early in learning indicate that in many episodes, the agent neither moves nor expects to move (Fig. 5(c)). Early dominance by the easy-to-observe *failure* suggests that it is inhibiting exploration.

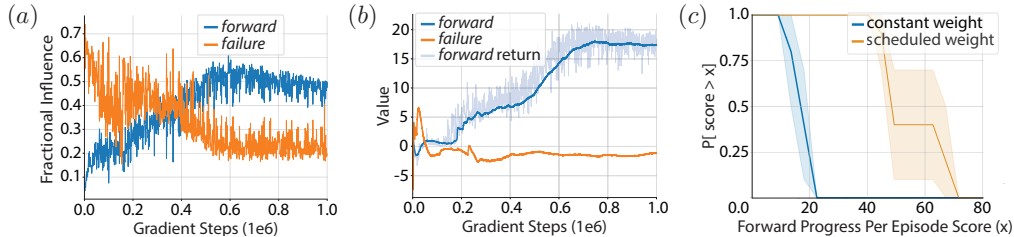

Figure 5: **Remedy for an adverse reward (a)** The initially low fractional influence of *forward* (blue curve) indicates it may be initially inhibited by failure penalties with higher influence. **(b)** The *forward* value predictions and returns (blue curves) are near zero early in training, indicating the agent does not expect to move forward. **(c)** An agent trained with its failure weight scheduled from a value close to zero up to one (orange curve) improves early forward progress compared to an agent with a constant failure weight (blue curve). Forward progress is measured as the average forward progress return from all evaluation episodes of the first 200k gradient steps. Shaded areas represent 95% bootstrap confidence intervals from 10 trials.

To mitigate this problem, we vary the *failure* component weight, $w_{\text{failure}}$, from 0.01 to 1 over the course of learning according to the schedule described in App. J. This schedule significantly increases the agents' *forward* progress (Fig. 5(c)), and accelerates learning.

## 7   Related work

Our work builds upon earlier studies of value decomposition in the explainable reinforcement learning (XRL) literature: DrQ [19], DrSARSA [29], and HRA [31].[9] Like our approach, these methods learn separate value function estimates for each term of a linear reward function. DrQ and HRA are off-policy Q-learning-like methods, while DrSARSA is an on-policy method. HRA does not converge to a globally optimal policy, as each value function is only *locally*-optimal for the reward component it measures. The RDX and MSX metrics proposed by DrQ could be adopted in our setting, but the influence metric is easier to use with continuous-actions, and aggregate with summary statistics. Our approach improves upon these prior contributions by: (1) working in continuous-action and discrete-action environments; (2) allowing for and demonstrating dynamic re-weighting of reward components during training; and (3) being applicable to a family of actor-critic methods. While these approaches and ours explore using value decomposition for explainability, these works focus on describing why an agent took certain actions to users, whereas we focus on how to use value decomposition to diagnose and remedy learning challenges.

The Horde architecture [34] and UVFA [30], methods for multi-goal learning, also employ multiple value functions (one for each goal). UVFAs use a parameterized continuous space of goals, while Horde makes multiple discrete value function predictions. The value functions in our work are conditioned on a policy that optimizes the global reward, whereas in Horde and UVFAs the value functions are conditioned on independent policies that greedily optimize local goals (similar to HRA). Additionally, in our approach, the composite value function can be recovered from the components. Other value decomposition work includes Empathic Q-learning [21] and Orchestrated Value Mapping [10]. The primary difference between our work these other approaches regards the motivation and application of value decomposition. These works focus on how value decomposition can improve sample efficiency, generalization, or other aspects of the core learning problem, rather than how to diagnose and remedy problems.

Mathematically, value decomposition bears resemblance to work on successor features, which has focused primarily on state representation and transfer learning [3, 4, 6, 12]. Methods for multi-objective RL [16, 28] learn sets of policies, each with a distinct linear weight over multiple reward objectives. Our work also recovers the value function for different linear reward preferences, but uses this capability to diagnose behavior and learning problems rather than to learn multiple policies.

---

[9]For a broad overview of XRL, we direct the reader to Heuillet et al. [17].

## 8 Concluding remarks

We have argued that the iterative design of reinforcement learning agents can be improved through the use of *value decomposition*, in which we keep individual reward components separate and learn value estimates of each. We provided a simple prescription for deriving value decomposition algorithms from actor-critic methods, and applied it to SAC to derive the SAC-D algorithm. Combined with the CAGrad method, SAC-D meets or exceeds SAC's performance in all environments we tested. We introduced the *influence* metric, and demonstrated its use in measuring each reward component's effect on an agent's decisions. Finally, we provided several examples of how value decomposition can diagnose and remedy agent learning problems.

Although value decomposition is a simple and broadly applicable tool, we note the following limitations. (1) Our method requires a composite reward function of multiple components. (2) We only study linear reward decomposition. (3) Component predictions only tell you the agent expectations under the single learned policy; changing the weights doesn't tell you what to expect after re-optimizing the policy for them. (4) Our approach benefits most with methods that learn Q-functions. Methods that optimize the policy with empirical returns have a weaker link between agent expectations (component Q-functions) and policy decisions. The influence metric also requires a Q-function model.

Our method presents the same societal benefits and risks as other RL methods. However, we believe this technology has a net beneficial impact, because making agent decisions more introspectable enables developers to catch problematic behavior before deploying the technology. One particular concern, however, is that value estimates represent an agent's beliefs, not ground truth; this should be kept in mind when such predictions are used in real-world decision-making, as they may reinforce biases or lead to incorrect conclusions.

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
