# OpenReview forum: "Value Function Decomposition for Iterative Design of Reinforcement Learning Agents"
_NeurIPS.cc/2022/Conference — NeurIPS 2022 Accept_

### Official Review · Reviewer_2638 · 2022-07-07

**Rating:** 3
**Confidence:** 4
**Soundness:** 1 poor
**Presentation:** 2 fair
**Contribution:** 2 fair

**Summary:**

The paper proposes to use value decomposition for the critic of generic actor-critic (AC) methods, and then applies it to soft-actor-critic (SAC) as a particular case. The idea of value decomposition is not new and has been studied in various prior work over past 20 years or so. Thus, the actual contribution of the paper is simply to use value decomposition in the context of AC methods.

**Questions:**

- The paper lacks any thorough analysis of possible issues and meaningful solutions with formal ground. This is essential.

- L127-128: "Because the component weights are factored out of the component Q-functions, they may be varied without changing the component prediction target."

This is not correct. First, changing the weights will change Q, hence, it may change j at line 4 of Algorithm 1. Second, changing Q induces a change of policy. Notice that your value learning is on-policy and changing Q can beaks that because the policy that collects the transition becomes different from the learning policy. At the very least this can make the learning process unstable/slow depending on how much and how fast the weights are changing. This is not a trivial issue. Note also that if Q-learning is used instead of SARSA to make it off-policy, then the entire algorithm will NOT converge to optimality due to Jensen’s inequality. See reference [3] bellow for a detailed analysis.

- L186: Not clear why norm is used (also not clear what norm it is). Is your action-space multi dimensional? If yes, you may need to describe it properly in your section 2.1.

Some references:

[1] van Seijen, H., Fatemi, M., Romoff, J. & Laroche, R. Separation of Concerns in Reinforcement Learning. arXiv [cs.LG] (2016)

[2] Fatemi, M. & Tavakoli, A. Orchestrated Value Mapping for Reinforcement Learning. in International Conference on Learning Representations (2022).

[3] Laroche, R., Fatemi, M., Romoff, J. & van Seijen, H. Multi-Advisor Reinforcement Learning. arXiv [cs.LG] (2017)


**Limitations:**

The authors counted four limitations, which I agree with in general. However, my rejection of the paper is not grounded on these limitations.

**Strengths And Weaknesses:**

Strengths:
Nothing in particular.

Weaknesses:
- As the main idea of the paper is quite simple, I expected a neat mathematical analysis. For example, what is the impact of value decomposition on AC methods and why they should be used? When and how they change bias/variance of the base AC method? Does using value decomposition influence convergence? etc...

- Re-weighting the reward components may make the algorithm completely unstable and even non-convergent. A clear analysis is required, beyond the few presented experimental results.

- The presentation may also be improved. This paper mostly looks like writing a report around some experimental results.

- Minor: some missing key references for value decomposition.

---

> ### Author Response · Authors · 2022-08-02
> **Response to comments and questions**
>
> We thank the reviewer for their feedback. We believe there was confusion in some of the issues they raised that we address below.
>
> Issue 1
> The reviewer expressed concern that we provided no theoretical guarantees for our algorithm.
>
> Typically, off-policy DRL algorithms do not have convergence guarantees except in the tabular setting. Our goal was not to use value decomposition to improve theoretical guarantees, but to provide more insight into the agent. Providing additional guarantees is well outside the scope of this work. However, if we restrict ourselves to the tabular settings, our approach retains these convergence guarantees.
>
> Our appendix includes a proof about how to recover the composite value function from component value functions. It does not require much work to show that the whole algorithm converges under the same assumptions of the underlying algorithm. Briefly, by substitution, we can first upper bound the error of the recovered composite value function by the maximum error of any single component and then show convergence from the convergence of the component Q-functions. Standard policy improvement theorems apply from there.
>
> We did not include this proof because it is not instructive of how value decomposition works. And since off-policy DRL typically does not have theoretical guarantees except in tabular settings, it is not an especially comforting result! However, if the reviewers think it would be useful, we will add this additional proof in the appendix.
>
> Issue 2
> The reviewer commented that re-weighting the reward weights may make the algorithm unstable.
>
> Changing the reward function does not break convergence if the reward function eventually stops changing. Policy iteration finds the optimal policy from any initial policy and policy evaluation converges from any value initialization. Ergo, if you eventually stop changing your reward function, the whole process converges. You can make an even stronger claim about other conditions, but in our work, we only ever proposed applications that eventually stop changing the reward function.
>
> Additionally, various kinds of non-stationary rewards have been explored throughout the literature (exploration bonuses, shaping rewards,etc.). We provide no other analysis of learning with dynamic rewards because there isn't anything new to add and because it's well explored in RL already. The novelty of what we did comes from the fact that with value decomposition we can evaluate the policy under any reward weights.
>
> Issue 3
> The reviewer claimed we were wrong about the fact that reward weights could be changed without changing the prediction target. The reviewer's objection was that because reward weightings change the policy, the value function must change too. We believe the reviewer may have misread the equation and text in question. We did *not* make any claim about the relationship between Q* for some weights and the component predictions. We made a claim about the relationship of Q^pi and the component predictions Q^pi_i. (Note the pi superscript in Equation 3.) That is, our claim is regarding policy evaluation. The weight decoupling we describe allows you to re-evaluate the current policy under different weight assignments.
>
> Being able to also instantly derive the optimal policy's value function for different weights would be a useful tool, but it is not what we claimed to do and the ability to re-evaluate the current policy under different weights without changing the target is still useful. In particular, we leverage the weight decoupling to make influence estimates.
>
> Issue 4
> The reviewer wondered how our work would extend to the off-policy setting with an algorithm that doesn't use SARSA. Under the common definition of “off-policy,” SAC is already an off-policy algorithm and it does not use SARSA for policy evaluation. That  is, where "off-policy" refers to algorithms that can evaluate and/or improve a policy using data collected by arbitrary policies. SAC is off-policy in this way. It's possible that by "off-policy" the reviewer meant "to use an algorithm that directly estimates Q*." However, that excludes all actor critic algorithms and our work focuses on value decomposition for actor critic algorithms.
>
> If the reviewer is interested in Q-learning-like methods with value decomposition, we direct them to the paper "Explainable reinforcement learning via reward decomposition," which we cited in our introduction. However, if they are interested in conventional off-policy learning with value decomposition, then they will be comforted to know that it already works with SAC.
>
> Issue 5
> The reviewer asked about the influence norm and action space.
>
> Yes, as we note in the introduction and the influence section, we focus on continuous action spaces, which are typically multidimensional. We will update the text to say "multi-dimensional continuous action spaces." We use an L2 norm as indicated in Equation 4 (we used ||.||_2 to indicate an L2 norm).

---

> > ### Comment · Reviewer_2638 · 2022-08-07
> > **Response to Author's Comments**
> >
> > I thank the authors for their response. As it stands, this paper is a clear reject for two reasons: lack of novelty and lack of reasoning about its workability (as it appears there are many cases that varying reward weights can break the convergence).
> >
> > While I am personally an advocate of value decomposition, the analysis of algorithms with value decomposition is far from trivial and their practicality and implementation requires extra care in various levels. I certainly expect a NeurIPS paper to address these, specifically since novelty is marginal in the case of this submission. It is unfortunate to see little meaningful response in the rebuttal.
> >
> > I would comment on the provided response. Hopefully these help:
> >
> > - It is easy to show that changing reward weights eradicates convergence altogether (the value function depends on weights, so as the policy grad). Settling down the weights if it happens slower than decaying of the learning rate, again eradicates convergence. On the other hand, settling them too early defeats the purpose and one can ask why not simply use the final weights in the first place? This is why you need to have a precise analysis and explanation. It is your choice, but surely studying the tabular case gives you a better understanding of what to expect when a nonlinear approximation is used; after all, tabular is a special case of linear, and linear is a special case of nonlinear. Clearly, a meaningful insight is missing here.
> >
> > "Our goal was not to use value decomposition to improve theoretical guarantees, but to provide more insight into the agent."
> > - Value decomposition (VD) has already been studied extensively in the literature, way beyond what is presented here (for a few samples see the references in my original comments). The main idea of this submission is quite simple and there is no novelty in it: The only point of this paper is to use VD in the critic of PG. At the very least you need to provide discussions on when and why it may not work at all, and the conditions under which the decomposition is acceptable in practice.
> >
> > on-policy vs. off-policy
> > - My comment was text-book basic and the point apparently is missed altogether. Your algorithm is on-policy because it uses the current policy for bootstrapping: line 2 of Algorithm1, $a'$ is sampled from the current policy not from data or optimizing by taking max. Thus, the value of the next state in your value loss may not match the value of the current state since they come from different policies. Note here that changing the reward weights can significantly exacerbates the mismatch (which does not exist in normal actor-critic algorithms). If there is a significant mismatch between your sampled $a'$ and the data from which $(s,a,s')$ is sampled, then you simply refer to a wrong evaluation in the policy grad when you use $Q(s',a')$, and the algorithm can easily become unstable. On the other hand, if you want to make it off-policy by using $max_{a'}$ rather than sampling $a'$, then there will be the *attractor* issue (see reference [3] in my original comments for extensive details). Again, that is why you need to have a clear analysis and explanation.
> >
> > Policy Iteration (PI)
> > - Not sure if I understand your argument about PI or if it is relevant. [if you'd like to relate the issue to PI, you can see that your evaluation step can be off due to wrong bootstrapping, hence the policy update is not necessarily a policy improvement step and the algorithm can remain in a fluctuating loop. The induction from PI's convergence is not correct].

---

> > > ### Author Response · Authors · 2022-08-09
> > > **Addressing remaining issues**
> > >
> > > We thank the reviewer for their subsequent response. While value decomposition can be studied from an algorithmic perspective, which we do think is a valuable approach, that is not our focus here. Instead we focus on demonstrating how value decomposition provides a tool for engineers to debug and design agents in an iterative manner. We believe this perspective is novel and has not been thoroughly explored elsewhere. That said, we believe that, from an algorithmic perspective, there are several misunderstandings with the critique given by the reviewer, which we have tried to explain below.
> > >
> > > One of the reviewer's points regards problems when value components are evaluating locally-optimal policies. The attractor issue from [3] regards this kind of evaluation setting. However, these problems do not exist in our setting. In our work, each value component Q_i^pi(s, a) answers “what is the expected component return starting in state s, taking action a, and then following the actor?” It does not answer “what is the expected component return starting in state s, taking action a, and then following the policy for maximizing reward component i alone?” The actor in our work then optimizes the global reward. In addition to our setting avoiding this problem, we believe the component values we learn are more useful when diagnosing and remedying global-reward maximizing agents, which is the goal of our work.
> > >
> > > While we might have made it more clear in our initial response, we did not intend to ignore the reviewer’s point about off-policy. We hoped to add clarity about the meaning of the term "off-policy" because it is commonly used to mean something other than how the reviewer used it and we don't want there to be confusion about the capabilities of our method. We use the same definition of “off-policy” used in Reinforcement Learning an Introduction Second Edition (page 103), and the same definition as the original papers for SAC, DDPG, and TD3 which all describe themselves as "off-policy." In this definition, an off-policy algorithm is one in which the target policy being learned about (evaluated and/or improved) and the behavior policy used to collect data are different. In SAC, the bootstrap target does not use the behavior policy's action; it is "off-policy" under this definition. Similarly, SAC-D can learn all its value components off-policy.
> > >
> > > Regarding Policy iteration's (PI) relevance, it is relevant because actor-critic algorithms are (loosely speaking) model-free generalizations of it. Actor-critic theory usually starts with PI and then generalizes. For example, the original policy gradient theorem: "Policy Gradient Methods for Reinforcement Learning with Function Approximation," starts with PI with function approximation  (section 4). The SAC paper also starts with PI. We are confident our method is using the correct bootstrapping target. The reviewer may be misunderstanding if they are assuming we’re doing something else (e.g., we are not evaluating locally-optimal policies, we are evaluating the actor model) or may have a misunderstanding of how SAC and similar algorithm’s off-policy policy evaluation works.
> > >
> > > As far as reward weight changing, it does not "eradicate convergence." As we described in our initial response, our work fits into a broad class of reward changes that do not remove any existing theoretical guarantees. It may be instructive to point out that in addition to that, SAC, by construction, includes non-stationary rewards because it includes a policy-entropy bonus and the policy changes with time. Nevertheless, SAC maintains convergence guarantees (again, in the tabular case which is the only space in which convergence guarantees and analysis for these kinds of algorithms exist). The broad claim that "changing rewards eradicates convergence guarantees" is not accurate in general, nor accurate for the specific construction we used.
> > >
> > > Regarding citations and novelty, while the work the reviewer cited is broadly related, and we will include it in the related work section with additional space of a final submission, these works do not explore the contributions we enumerated. We also disagree that "the only point of this paper is to use VD in the critic of PG." While the reviewer is entitled to the opinion that our contributions are unimportant (though of course we disagree), these cited works are not evidence that our investigation and proposals have been done previously. This paper retains its value in light of them because they do not address the issues we tackle.
> > >
> > > For providing a discussion of when value decomposition works, one of our contributions is studying how and why adding value decomposition naively can reduce performance in various environments. We additionally proposed methods to address these issues that re-gains the performance of the underlying algorithm and sometimes exceeds it. Sections 3.2, 4, and additional material in the appendix, are dedicated to this investigation.

---

### Official Review · Reviewer_cBJt · 2022-07-11

**Rating:** 7
**Confidence:** 3
**Soundness:** 3 good
**Presentation:** 3 good
**Contribution:** 3 good

**Summary:**

This paper proposes a tool to diagnose and design the reward function for training the reinforcement learning (RL) agent. In practice, the reward function in RL consists of multiple terms (e.g., collision penalty and velocity bonus). The proposed method assumes the reward is the linear combination of these terms and learn a Q-function for each reward term. Such a decomposition enables the reward function designer to inspect the influence of each reward on the agent's decision, thus reducing the efforts of tuning the reward function without guidance. Several diagnosis case studies showcase how RL practitioners use this tool to design the reward function.

**Questions:**

- The current formulation needs the reward function to be a linear combination of multiple rewards. Can the author comment possibility of extending the proposed method to non-linear rewards?
- In section 6, the results of some diagnosis (e.g., weight scheduling, markov features) can be applied to SAC without value decomposition. Have the authors tried applying weight scheduling and markov features to SAC without value decomposition? This might help answer whether the diagnosis results can be applied back to non-decomposed version.

**Limitations:**

The author acknowledged the limitations in the paper.

**Strengths And Weaknesses:**

Originality:

Though value decomposition is not a new idea, using value decomposition to guide the reward function design and the proposed influence metric are an original contribution, to my best knowledge.

Quality and clarity:

This paper is easy-to-follow. The experiments are well-designed.

Significance:

The proposed reward design scheme is important in RL.

---

> ### Author Response · Authors · 2022-08-02
> **Response to comments and questions**
>
> We thank the reviewer for the comments and questions. We respond to each of their questions below.
>
> Q1: In our experience, the linearity of the reward decomposition has not been a major limitation, and thus far, we have not encountered environments in which extending to the non-linear composite reward function would be useful. One reason why we haven't seen environments that would benefit from nonlinear composite reward functions is because, while the composite reward function must be linear over the reward components, each reward component may be a nonlinear function of state variables. Consequently, it's often natural to represent rewards as linear functions of non-linear reward components. However, if we had more examples of such environments, it would be a fascinating space to explore!
>
> Speaking to that, we imagine the core challenge with nonlinear composite reward functions is how to recover the composite value function from component value predictions. Linear composite reward functions have the convenient property that the linear relationship between composite reward and its components is the same as the relationship between composite value function and the component value functions. Consequently, it is straightforward to recover a composite value prediction from the component value predictions. In contrast, if the composite reward is a nonlinear function of its components, the relationship between the composite value function and the component value predictions is not necessarily the same.
>
> If it could be shown that there is a fixed relationship between the composite value function and component value functions for some nonlinear reward function, then our approach would generalize straightforwardly to nonlinear reward functions. If the relationship is unclear (which we suspect to be the case for most non-linear reward functions), one way this issue might be tackled in the future is to let a neural network learn the composite value function relationship to the component value functions. Without exemplar environments where this would be useful, however, we haven't explored this direction.
>
> Q2: We agree that after diagnosing the problem with value decomposition, some (but not all) of the methods we use to remedy the problem could be applied without value decomposition. This could lead to an iterative design pattern where initially agents are trained with value decomposition to diagnose problems, and then subsequently trained without decomposition when remedies have been applied. The inclusion of Markov features is an excellent example of a remedy that could be applied in either case after the problem was diagnosed with value decomposition. However, given that our results indicate that policy performance when learning with value decomposition is comparable to performance when learning without decomposition, we find that training without decomposition is an unnecessary extra step.
>
> Additionally, some of the remedies we explored cannot be done exactly the same without value decomposition. The weight scheduling, for example, cannot be replicated exactly without value function decomposition. The closest analog in vanilla learning would be to dynamically change the reward signals with a schedule. While we suspect that would still be helpful, note that it has some key differences. If you change the reward signal itself, then value predictions must be adapted even if the policy doesn't change. If you instead change the decomposition weights (decoupled from the component signal), as we did, then the value predictions only need to adapt if the policy changes. This property may be particularly useful if the learned policy is already doing well before the weights are finished changing. For example, increasing a fail penalty weight to a larger value may not change the policy much if the agent is already doing well, but changing the reward signal itself may significantly change the value prediction magnitude to account for the rare events when the agent fails and incurs a large cost.

---

### Official Review · Reviewer_yA8d · 2022-07-11

**Rating:** 7
**Confidence:** 3
**Soundness:** 4 excellent
**Presentation:** 4 excellent
**Contribution:** 3 good

**Summary:**

This paper proposes *value decomposition* for diagnosing and iteratively designing actor-critic algorithms. The key idea of value decomposition is to treat the reward signal as constituted by a number of distinct components that together yield value. However, rather than just take some simple operation of these distinct signals, value decomposition proposes to learn a $Q$ function for each separate component of the reward: a Q-network, then, outputs $m$ predictions (for $m$ reward components) rather than just $1$. This relatively simple idea is explored as a mechanism for diagnosing a variety of characteristics of an RL algorithm, and ultimately give rise to the design and improvement of new algorithms. As the paper points out, the core idea of value decomposition is not new, and has been applied to Q-learning variants in the past. Indeed, value decomposition is conceptually similar to typical linear assumptions on the reward or value, but encourages algorithm design to explicitly learn these distinct value components. Moreover, to my knowledge, this is the first extension to actor critic algorithms: the paper proposes SAC-D, a version of Soft Actor-Critic with value decomposition in the mix. Section 4 presents findings from experiments contrasting various forms of SAC-D in different continuous Gym environments, such as Bipedal walker or Ant.

Arguably the most significant aspect of value decomposition is that it unlocks new perspectives for diagnosing agent failures, and for iteratively improving the design of these agents in light of these failures. Section 5 presents a series of experiments and insights showcasing how dropping out one of the reward components can impact aspects of the agent. Figure 2, for example, illustrates the the influence each reward component exerts on the Q function through contour maps over different mixes of the reward components $Q_1, Q_2$ and $Q_3$, while Figure 3 explores the _fractional influence_ of each component. I found Figure 3b to be particularly illuminating. These insights culminate in Section 6 that provides a pragmatic view on how to incrementally improve agent design using the diagnostic tool of value decomposition. For instance, in Lunar Lander, certain reward components should be negative, while others should be non-negative. Observing that an agent is incorrectly predicting certain values to be outside their possible bound allows for incremental adjustment of the agent.

**Questions:**

Q1: At first I found the extension to SAC quite natural, but reading it through a few times, I find it less clear why SAC is needed for the rest of the story. To me the main claim of the work is that value decomposition can be a powerful diagnostic tool for understanding agent limitations, and fixing them. This perspective applies just as easily to the existing Q-learning approaches to value decomposition. So, I am left wondering: why is SAC the focus, rather than Q-learning? Another way to put it: do you anticipate that the pragmatic suggestions from Section 5 and 6 extend to Q-learning based value decomposition as well?

Q2: Do you anticipate the linearity of the reward composition to be problematic or overly limiting?

Q3: One reaction I have to the work is that linear functions (and approaches focused on linearity) in general are praised for their interpretability. In this way, the idea of value decomposition is very nearly just making a linearity assumption. How, in general, should we think about this approach as different from linear RL, or just assuming that the reward admits a linear decomposition with some unknown weights as in Jin et al. (2020)?

Jin, Chi, et al. "Provably efficient reinforcement learning with linear function approximation." Conference on Learning Theory. PMLR, 2020.

**Strengths And Weaknesses:**

[STRENGTHS]

Overall, this paper proposes a novel perspective for diagnosing agent failures and iteratively improving agents as a result. As the introduction of the paper points out, this new viewpoint directly engages with the practical difficulties of deploying RL agents: they often fail for unknown reasons, and staring at a monolithic neural network and chaotic learning curves can often be daunting to debug and react to. The method of value decomposition is simple, easy to understand, and can be broadly applicable. The paper is well written and takes care in motivating many of its claims. It also simply lays out its core contributions, and limitations. The experiments are broad, interesting, and reveal the considerable potential behind the proposed method.

I believe this paper possesses many virtues, and is sure to be of interest to the community. For this reason, pending any issues uncovered by other reviewers, I recommend accepting this paper.

[WEAKNESSES]

There are several small things that could improve the paper, but none are major. First, the scope of the paper is quite ambitious, but I believe ultimately that sections 5 and 6 and the perspectives therein really constitute the core contributions. In this sense, it is not clear the new algorithm SAC-D or the experiments in Figure 1 are strictly necessary to the core of the work. A mild suggestion, but perhaps they can be moved to the appendix. Otherwise, I have a few minor writing suggestions and typos to fix (see below).

Writing Suggestions:
- In Algorithm 1, consider using the \texttt{} wrapper for the boolean "use_cagrad"
- I believe in Equation 2b $Q_i$ should be changed to $Q_j$.
- The pseudocode for Algorithm 1 might be cleaned up. For instance, I find the added text to detract from the readability of the algorithm. There are a few too many comments as well.
- Another minor point, but I found the y-axis of Figure 1b to be counter intuitive. The first few times I read this plot, I assumed the "Normalized Expected Reward" in Figure 1a was applying its y-axis to both plots. I wonder if there is some visual adjustment to be made that can prevent this collision.
- Small point of consistency: the subfigure labels differ throughout the paper in style.

---

> ### Author Response · Authors · 2022-08-02
> **Response to comments and questions**
>
> We thank the reviewer for their suggested writing improvements and comments. We will incorporate these suggestions into the final submission. Below are answers to the reviewer's questions.
>
> Q1: The reviewer is correct that SAC is not essential to our work. Indeed, our goal is that others could straightforwardly apply our extensions to other algorithms if they preferred a different algorithm! CAGrad should be useful in other value decomposition settings (including Q-learning settings) because it addresses a problem inherent to value decomposition itself: how to optimize multiple predictions simultaneously. There are also various actor-critic methods that use twin network minimums and all would benefit from how we adapt that to the value decomposition setting.
>
> The motivation for focusing on actor-critic algorithms is that they permit both discrete and continuous action spaces, whereas Q-learning methods only support discrete action spaces. We believe this action-space generality is important for greater adoption of value decomposition. While many actor-critic algorithms could have been used, we focused only on one (SAC) and validated that our extensions kept its performance competitive with the performance without value decomposition so that the remainder of the paper could focus on how to use value decomposition. We chose SAC because it's well studied in continuous action spaces, popular, and has state-of-the-art performance.
>
> Q2: We do not believe reward linearity is especially prohibitive. Reward functions are very often already designed as linear combinations of components. For example, the Gym implementations of the benchmark environment reward functions were already linear combinations of reward components; we didn't have to do anything creative.
>
> It is worth highlighting that the linearity requirement is *not* a requirement that the reward is a linear function of *state variables.* Only that is a linear function of some components which may be non-linear functions of state variables themselves. For example, one of the shaping components in lunar lander is a Euclidean distance of position: a nonlinear function of the state variables. We will highlight this point more in the final submission.
>
> Q3: The primary difference between our approach and linear RL is linear RL requires linearity of the value function and policy over the *state representation*. Linear RL is quite limiting because even simple reward functions often induce complex value functions and optimal policies that cannot be practically estimated with linear functions and demand nonlinear function approximation. Fortunately, we do not require the value function to be linear over states, only the reward function to be linear over some set of components. In all our examples, we used non-linear MLPs so that each of the $m$ value component predictions is a nonlinear function of state. We then recover the composite value function by a linear combination of these non-linear value component predictions.

---

> > ### Comment · Reviewer_yA8d · 2022-08-05
> > **Response to Author's Comments**
> >
> > I thank the authors for their thorough response to my review and questions. The answers provided definitely help.
> >
> > I maintain a positive outlook on the paper: I believe the perspectives provided here on value decomposition can be generally of interest to the community.

---

### Official Review · Reviewer_Ssv2 · 2022-07-11

**Rating:** 4
**Confidence:** 5
**Soundness:** 2 fair
**Presentation:** 3 good
**Contribution:** 2 fair

**Summary:**

This paper provides a framework to integrate value decomposition into a broad class of actor-critic algorithms, and applied it to SAC to derive the SAC-D algorithm. Lots of experiments are conduct to verify SAC-D's performance over SAC. This paper introduces the influence metric, and demonstrated its use in measuring each reward component's effect on an agent's decisions. Finally, they provides several examples of how value decomposition can diagnose and remedy agent learning problems.

**Questions:**

(1) Since SAC-D is developed by incremental, heuristic refinement  to SAC, it is not clear whether the performance gain obtained by this refinement is limited to a few experiments in this paper.
(2) Value decomposition in this paper is artificially divided. How does value decomposition with different divisions affect the effect of the algorithm?

**Limitations:**

Yes.

**Strengths And Weaknesses:**

Originality:
The operation of integrating value decomposition into reinforcement learning algorithms seems to be common sense, which has been proposed by many existing literature. And the performance of value decomposition has been explored in a large body of literature. The work of this paper seems to be just another confirmation of previous findings. Therefore, this paper is less original.

Quality:
This paper is well written and easy to understand. But, I have two concerns as follows. (1) Since SAC-D is developed by incremental, heuristic refinement  to SAC, it is not clear whether the performance gain obtained by this refinement is limited to a few experiments in this paper. (2) The iterative design of RL agents shown in Section 6 is quite interesting. However, it still seems to require strong task-specific priors from experts.

Clarity:
This paper is well written and easy to understand.

Significance:
The work is a bit significant, but does not give new insights into value decomposition.

---

> ### Author Response · Authors · 2022-08-02
> **Response to comments and questions.**
>
> We thank the reviewer for their thoughtful comments, and we agree that value decomposition is common sense! Indeed, a major aim of our paper is to provide a framework to formalize its broad applicability and to illustrate its use in practice. This is the context in which we think about our performance comparisons and our iterative-design examples.
>
> We agree with the reviewer that further evidence would be required to argue that SAC-D outperforms SAC in any general sense. However, our goal in the robustness experiments is not to argue that SAC-D outperforms SAC, but to argue that the two models are comparable in policy performance so that learning dynamics can be studied (using the value decomposition methods described in the paper) without a compromise in policy performance. With the additional space of the final submission, we will add text to the results section to better contextualize the results under this goal.
>
> Regarding originality, while there exists other value decomposition work, to our knowledge, this is the first time such a framework has been proposed in the value decomposition literature. In particular, again to the best of our knowledge:
> 	- Previous work has not studied value-decomposition for continuous action-space problems.
> 	- Our analysis of learning problems that arise when using value decomposition is novel. (Although we also propose solutions to these problems -- e.g., using CAGrad -- our identification of the problems lays the groundwork for further research.)
> 	- The framework from which we derive SAC-D is novel (and should be applicable to other continuous action-space RL algorithms – e.g. DDPG, TD3).
> 	- The influence metric (and its use to diagnose reward component contribution to decision-making) is novel.
> 	- We are not aware of any previous illustrations of value decomposition’s use in iterative design.
>
> Regarding reward component choice, we find that designing rewards as linear functions of components is already a natural practice. For example, we did not artificially choose reward components for the benchmark environments. Rather, the existing Gym implementations already implemented the reward as a linear function of components. We simply exposed those already-existing components.

---

> > ### Comment · Reviewer_Ssv2 · 2022-08-06
> > **The reviewer's response**
> >
> > I'm still concerned about the lack of originality of this paper. It doesn't seem to bring new insights to the value decomposition community.

---

### Meta-Review · Area_Chair_bbgo · 2022-08-26

**Recommendation:** Accept
**Confidence:** Less certain

**Metareview:**

The reviewers carefully analyzed this work and agreed that the topics investigated in this paper are important and relevant to the field. However, reviewers expressed different opinions on the merits and contributions of this work.

One reviewer pointed out that similar ideas have been explored before and that the paper does not necessarily give new insights into value decomposition. The authors counter-argued by saying that the proposed method is the first concrete value-decomposition framework (with these particular properties) to be proposed in the literature.

Another reviewer had a more pessimistic view of the merits of this work. They argued that the paper lacks novelty and simply applies the idea of value decomposition to a different setting—in the context of actor-critic (AC) methods. They also argued that they expected a more formal discussion about the impact of value decomposition on AC methods and on whether re-weighting reward components could cause instability. The authors responded that off-policy DRL algorithms typically do not have convergence guarantees and that their goal was orthogonal: to provide more insight into the design of agents. The authors also argued that standard convergence guarantees apply if the reward function stops changing. Ultimately, this reviewer's two main points of contention were:
1. From their point of view, methods that help implement agents via iterative design could, in principle, be interesting for discussion in other venues, such as a workshop paper, but that this level of contribution does not meet the bar for a full conference paper; and
2. Having dynamic weights could cause the method to become unstable and the paper did not study whether this could be a problem in practice.

Two other reviewers, by contrast, expressed very positive views on this work.

One reviewer argued that even though the general idea has been explored before, using it to guide the reward function design is an original contribution. They also argued that the experiments were well-designed. After reading the authors' rebuttal, this particular reviewer said that the authors addressed all their technical questions.

Another reviewer expressed similarly strong positive views on this paper, arguing that:
1. It explores an important direction that ultimately leads to the design and improvement of new algorithms.
2. Even if the underlying idea is not new, this is the first extension of that idea to actor-critic algorithms.
3. The paper proposes a novel perspective for diagnosing agent failures and iteratively improving agents, and that this new viewpoint directly engages with the practical difficulties of deploying RL agents.
4. This reviewer further supported their positive view of this paper by stating that the experiments are broad, interesting, and reveal the considerable potential behind the proposed method. For these reasons, they believe this paper will likely be of general interest to the community.

Ultimately, the main point of contention between the opinion of these two reviewers seems to result from their opposite views on what, precisely, is the contribution of this work:
- One reviewer sees this paper as one that merely applies the idea of value decomposition of actor-critic methods, without providing thorough theoretical and formal analyses. This reviewer argued that this contribution does not meet the bar for publication as a full conference paper.
- The other reviewer views this paper as one that introduces novel diagnostic tools for understanding learning agents. They argued that they found the cleanliness around the proposed diagnostic tools to be appealing and that the resulting proposal for iterative design, based on these diagnostics, is broadly useful for RL. This reviewer highlighted three kinds of diagnostic unlocked by the perspective introduced in the paper: **(1)** identifying insufficient state features; **(2)** value prediction errors; and **(3)** reward and exploration. They believe these avenues are new and useful when viewed from the perspective of iteratively identifying and correcting failure modes of agents. Ultimately, they disagreed with the reviewer above by arguing that they did not find the attachment to actor-critic algorithms all that critical.

Overall, thus, it seems like there is disagreement regarding the merits of this paper primarily due to two reviewers not agreeing on whether the proposed iterative design method is (in principle) a sufficient contribution to a conference paper. Regarding this matter, a fourth reviewer expressed a positive view of this work by stating that the questions studied here address an important aspect missing in the literature: how to diagnose agents. They argued that—given this paper's contributions towards that goal—the lack of formal analyses should not be considered a critical reason for rejection.

All reviewers encourage the authors to update their work based on their constructive criticisms and, in particular, in a way that tackles the points of contention mentioned in the original reviews and their post-rebuttal comments.

**Award:**

No

---

### Decision · Program_Chairs · 2022-09-14

Accept